# Electron Paramagnetic Resonance as a Tool for Studying Membrane Proteins

**DOI:** 10.3390/biom10050763

**Published:** 2020-05-13

**Authors:** Indra D. Sahu, Gary A. Lorigan

**Affiliations:** 1Natural Science Division, Campbellsville University, Campbellsville, KY 42718, USA; 2Department of Chemistry and Biochemistry, Miami University, Oxford, OH 45056, USA

**Keywords:** membrane protein, electron paramagnetic resonance (EPR), site-directed spin labeling, structural and dynamics, membrane mimetic, double electron electron resonance (DEER)

## Abstract

Membrane proteins possess a variety of functions essential to the survival of organisms. However, due to their inherent hydrophobic nature, it is extremely difficult to probe the structure and dynamic properties of membrane proteins using traditional biophysical techniques, particularly in their native environments. Electron paramagnetic resonance (EPR) spectroscopy in combination with site-directed spin labeling (SDSL) is a very powerful and rapidly growing biophysical technique to study pertinent structural and dynamic properties of membrane proteins with no size restrictions. In this review, we will briefly discuss the most commonly used EPR techniques and their recent applications for answering structure and conformational dynamics related questions of important membrane protein systems.

## 1. Introduction

### Membrane Protein

Understanding the basic characteristics of a membrane protein is very important to knowing its biological significance. Membrane proteins can be categorized into integral (intrinsic) and peripheral (extrinsic) membrane proteins based on the nature of their interactions with cellular membranes [1]. Integral membrane proteins have one or more segments that are embedded in the phospholipid bilayer via their hydrophobic sidechain interactions with the acyl chain of the membrane phospholipids. Integral membrane proteins spanning the width of the lipid bilayer are known as transmembrane proteins. Membrane-spanning domains of transmembrane proteins are mostly α-helices or multiple β strands. Peripheral membrane proteins do not interact with the hydrophobic core of the phospholipid bilayer. Instead, they are bound to the membrane indirectly by interactions with integral membrane proteins or directly by interactions with polar lipid head groups. Peripheral proteins are usually localized at or near the cytosolic face of the membrane [1].

Membrane proteins are involved in many important biological functions for the survival of living organism. In humans, 30% of the genome encodes membrane proteins [2,3,4]. Genetic mutation and misfolding of membrane proteins are linked to numerous human dysfunctions, disorders, and diseases [5,6]. Membrane proteins are targets of more than 50% of total modern food and drug administration (FDA) approved drugs [7,8]. They also play a very important role in the development of antiviral and antibacterial agents [9,10]. The knowledge of structural dynamics and functions of membrane proteins is of high biological importance [11,12,13]. Membrane proteins contribute less than 2% of the structure in the protein data bank (PDB) [11,14,15]. Despite their abundance and importance, very limited information about membrane proteins exists when compared to globular proteins due to challenges in applying biophysical techniques for studying these protein systems [16,17]. 

In recent years, several biophysical techniques have been utilized to investigate the structural and dynamic properties of membrane proteins. The most popular biophysical techniques are X-ray crystallography, nuclear magnetic resonance (NMR), electron microscopy (Cryo-EM), Förster resonance energy transfer (FRET), and electron paramagnetic resonance (EPR) spectroscopy [13,18,19,20,21,22,23]. X-ray crystallography is used to determine the highly resolved 3D structure of membrane proteins [24]. However, it is difficult for X-ray crystallography to reveal dynamic information of most of the proteins in a membrane. In addition, membrane proteins are difficult to crystallize as they are solubilized in detergent/lipids and have high hydrophobicity [18,20]. This introduces challenges for X-ray crystallographic techniques for studying many membrane proteins [25]. Nuclear magnetic resonance spectroscopy (NMR) is used to obtain both structural and dynamics information of a variety of membrane proteins in a non-crystal environment. However, size restriction is a major drawback in solution NMR spectroscopy (restricted to <50 kDa) [25,26,27,28]. Furthermore, NMR requires isotropic samples to avoid line broadening effects that introduce challenges in studying membrane proteins in proteoliposomes, which are the closest membrane mimetic of the cellular membrane environment. Cryo-EM is a very powerful and rapidly growing technique that has been used in recent years to investigate the structure of biological systems due to technical developments in instrumentation and sample preparation [29,30,31,32]. However, analyzing small proteins is challenging due to low resolution [33]. FRET is a good technique to monitor the conformational changes for individual membrane protein systems. However, this technique may cause higher structural perturbation due to the presence of relatively larger probe sizes. Furthermore, the site-specific incorporation of the probe throughout the sequence is also very challenging [20]. EPR spectroscopy is a powerful biophysical technique that minimizes these limitations and provides pertinent structural and dynamic information about membrane proteins. 

## 2. Electron Paramagnetic Resonance

Electron paramagnetic resonance (EPR) spectroscopy is a very important biophysical technique for investigating paramagnetic species, including organic and inorganic radicals, and paramagnetic triplet states. The working principle of EPR is similar to the working principle of NMR. The difference is that EPR focuses on the interaction of an external magnetic field with an unpaired electron spin in a molecule while NMR focuses on the interaction of an external magnetic field with isotopic nuclei of the individual atom. EPR measures an absorption of microwave radiation by an unpaired electron spin in the presence of an external magnetic field. The degeneracy of the unpaired electron is lifted in the presence of the external magnetic field resulting in two energy states, M_s_ = +1/2 and M_s_ = −1/2, where the lower energy state M_s_ = −1/2 is aligned parallel to the magnetic field and the upper energy state M_s_ = +1/2 is aligned antiparallel to the magnetic field. The energy difference between the lower and the upper state is given by
Δ*E* = *g_e_ βe B*_0_(1)
where *B*_0_ is the magnetic field, *g*_e_ is the electron’s so-called g-factor, which varies depending on the electronic configuration of the radical or ion, and *βe* is the electron Bohr magneton. Equation (1) implies that the splitting of the energy levels is proportional to the strength of the static magnetic field strength (*B*_0_) as shown in Figure 1 [19]. An unpaired electron spin can flip between the two energy levels by absorbing microwave radiation of energy *h*ν, obeying the fundamental equation of EPR spectroscopy [34].
Δ*E* = *hν*=*g_e_βB*_0_(2)
where h is Planck’s constant and ν is the frequency of the microwave radiation. Most EPR studies are conducted at 9 GHz (~3300 G), also known as the ‘X-band’ frequency. Q-Band frequency (~35 GHz) experiments are also relatively common.

### 2.1. Continuous Wave Electron Paramagnetic Resonance (CW-EPR) Spectroscopy

In a CW-EPR experiment, CW EPR spectra are collected by placing a sample into a microwave field of constant frequency, *ν*, and varying the external magnetic field, B_0_, until the resonance condition is satisfied. In the experimental setup, the microwave field is built in a resonator, where the sample tube is introduced. The resonator is critically coupled, which means that the incident power is completely absorbed by the resonator. Additional absorption by the sample during resonance leads to a detuning of the resonator and reflection of microwave power. The measurement of this reflected microwave power as a function of the magnetic field yields the corresponding CW-EPR spectrum (Figure 1). Amplitude modulation of the magnetic field with a frequency of typically 100 kHz increases the signal-to-noise (S/N) ratio considerably and is responsible for the derivative shape of the spectra. The excitation bandwidth of the microwave (MW) irradiation is very small (approximately 2 MHz) and the MW power is at maximum 200 mW. CW-EPR spectroscopy usually suffers from limited spectral and time resolution. 

### 2.2. Pulse Electron Paramagnetic Resonance Spectroscopy

In a pulse EPR experiment, the magnetic field is kept fixed and the EPR spectrum is recorded by exciting a large frequency range simultaneously with a single high-power MW pulse at a given frequency, *ν*. A pulsed experiment can provide isolation, detection, and measurement of the interactions that contribute to the shape and behavior of a CW spectrum. Since the relaxation times are too short for most of the biological systems at room temperature, pulsed EPR measurements usually require cryogenic temperatures. The most commonly used pulse EPR experiments are echo-detected field sweeps, double electron electron resonance (DEER), electron–nuclear double resonance (ENDOR), electron spin echo envelope modulation (ESEEM), and hyperfine sublevel correlation (HYSCORE). 

For details of the theory behind these EPR approaches, we refer to the following excellent references [35,36,37]. In the sections below, we will briefly introduce the EPR techniques commonly used for the study of membrane proteins. 

## 3. Biological EPR

In the past, the application of the biological EPR was restricted to metalloproteins possessing paramagnetic centers or enzymes with radical cofactors. The absence of unpaired electrons in most biological systems would appear to minimize the application of EPR methods. The development of molecular biology techniques incorporating stable radicals at specific locations on biological systems extended the application of EPR spectroscopy to nearly any biological system. The site-specific introduction of unpaired electrons into biomolecules in the form of spin labels is known as site-directed spin labeling (SDSL) [38,39]. 

### Nitroxide Based Spin Labeling EPR

In nitroxide based site-directed spin-labeling experiments, all native non-disulfide-bonded cysteines are replaced by another amino acid such as an alanine or serine. Site-directed mutagenesis is used to introduce a unique cysteine residue into a recombinant protein. The protein containing a site-specific cysteine is further reacted with a sulfhydryl-specific nitroxide reagent to generate a stable EPR-active spin-label side-chain [39,40,41]. 

Figure 2 shows the chemical structure of some nitroxide-based spin-label probes used for EPR spectroscopic studies of membrane proteins [13,27,42,43,44,45,46,47,48,49,50]. The spin-label probes in Figure 2 are incorporated using site-directed mutagenesis. A resulting side-chain produced by reaction of the most commonly used spin label, methanethiosulfonate spin label (MTSL), with the cysteine residue (T58C) of the KCNE1 membrane protein is shown in Figure 3 [13,51].

## 4. Nitroxide Based Site-Directed Spin Labeling EPR for Studying Membrane Proteins

Nitroxide based site-directed spin labeling EPR spectroscopy has been widely used to study membrane proteins. This is a very broad topic. In the following sections, we will discuss it in an introductory fashion with recent examples. For more in-depth information, we refer the following excellent reviews [12,13,42,47,50,54,55,56,57,58]. 

### 4.1. SDSL CW-EPR for Studying Structural Topology and Dynamic Properties of Membrane Proteins

The dynamic information about the spin-labeled side-chain of a biological system can be obtained by lineshape analysis of the corresponding CW-EPR spectra [12]. The flexibility of the MTSL nitroxide spin label provides its motion, which is highly dependent on neighboring amino acid side chains and secondary structure components in its immediate environment. The CW-EPR spectra are highly sensitive to the spin-label motion. The EPR spectral lineshape reflects the mobility of the spin-label side-chain and its relation to the structure and environment within the protein. The EPR spectrum for the spin labels moving rapidly in solution reduces to three isotropic peaks (Figure 4A). The spectrum is in the rigid limit when the spin-label motion is very slow such that it is close to motionless [59]. In the rigid limit, the sample is frozen and the full orientation-dependent parameters are observed. When the spin-label motion falls between these two regions, the dynamic properties of the site-specific spin label can be obtained by determining the rotational correlation time (τ_c_) [59]. The overall mobility of the spin label attached to the protein is the superposition of various kinds of motion including the motion of the label relative to the protein backbone, fluctuations of the α-carbon backbone, and the rotational motion of the entire protein. These motions can be separated from the EPR spectrum under different experimental conditions. A relative mobility of the spin label can be determined by calculating the inverse central linewidth of the EPR spectrum [12,51,60,61]. The binding properties of the protein/peptide and membrane can be investigated by measuring the changes in spin-label mobility [55,62]. In the aqueous phase, a spin-labeled peptide or a rapidly tumbling small protein leads to an isotropic spectrum with a rotational correlation time of less than 1 ns. However, in a membrane environment, the mobility of the spin-labeled protein is reduced, leading to a broader EPR spectrum with two motional components resulted from the superposition of the signals arising from a free and bound peptide [51,55,63,64,65,66]. The more quantitative information about the spin-label side-chain dynamics can be obtained with EPR spectral simulation approaches using freely available simulation programs such as Easyspin and non-linear least squares (NLSL) [59,67,68]. Figure 4 shows an illustration of CW-EPR spectra for the MTSL nitroxide spin label attached to KCNE1 reconstituted in different dynamic environments [69]. 

Nitroxide-based SDSL EPR power saturation experiments can be used to study the topology of the protein with respect to the membrane [13,57,64,65,70]. There are several biologically important protein systems such as *Escherichia coli* ferric citrate transporter FecA, vimentin, GM2 activator protein, ABC cassette transporter MsbA, cytochrome C oxidase subunit IV (COX IV), the prokaryotic potassium channel KcsA, KCNQ1-VSD, Pinholin, KCNE1, lactose permease protein, integrin β_1a_, functional amyloid Obr2A, C99 domain of the amyloid precursor protein, bacteriorhodopsin, KvAP voltage-sensing domain and phospholamban (PLB), and the GTPase domain of HydF that have been studied using nitroxide-based SDSL CW-EPR spectroscopy to probe the structural, topology, and dynamic properties [51,60,62,65,66,70,71,72,73,74,75,76,77,78,79,80,81,82,83,84,85].

A recent example of using site-directed spin labeling CW-EPR spectroscopy is the study of human KCNQ1-VSD in proteoliposomes [84]. The human KCNQ1 (Q1) is a voltage-gated potassium channel expressed in several tissues of the body and is known to regulate various physiological functions. It is a six-pass transmembrane protein involved in the repolarization phase of cardiac action potentials and was identified as the gene causing chromosome 11-linked Long QT syndrome [3,4,5,6]. Dysfunction of the channel has also been linked to other disease conditions like Romano–Ward syndrome, sudden infant death syndrome, congenital deafness, and familial atrial fibrillation [7]. The isolated-VSD domains can fold even in the absence of the pore domain (PD), suggesting that VSDs can adopt native-like structure independently of the PD [16,17,18]. CW-EPR power saturation data obtained on 20 sites of spin-labeled KCNQ1-VSD were used to determine the topology of KCNQ1-VSD with respect to the 1-palmitoyl-2-oleoyl-phosphatidylcholine (POPC)/1-palmitoyl-2-oleoyl-phosphatidylglycerol (POPG) lipid bilayers. Also, the data showed that all four transmembrane domains (S1–S4) are buried into the lipid bilayer, while the helix S0 of KCNQ1-VSD is solvent-exposed with some of the portions partially or weekly interacting with the membrane surface. Additionally, the CW-EPR lineshape analysis performed on 18 sites of spin-labeled KCNQ1-VSD suggested an overall restricted motion of spin-labeled Q1-VSD in lipid-bilayered vesicles when compared to that in the detergent micelles. This study further put together a structural topology model of KCNQ1-VSD in lipid bilayers. Figure 5 shows the proposed topology and the power saturation data on KCNQ1-VSD in lipid bilayers [84]. The CW-EPR power saturation data were analyzed to obtain peak-to-peak amplitude of the first derivative mI = 0 resonance line and to plot against the square root of the incident microwave power for three sample conditions: (1) equilibrated with nitrogen as a control; (2) equilibrated with lipid-soluble paramagnetic reagent 20% oxygen (air); and (3) equilibrated with nitrogen in the presence of a water-soluble paramagnetic reagent NiEDDA. Figure 5B,C show that residues Q147C and F222C of Q1-VSD have greater accessibility to NiEDDA, while F130C and F232C appear to interact more with O_2_ in the nonpolar lipid environment. The membrane depth parameter (*ϕ*) obtained from CW-EPR power saturation data (using Equation (3)) were plotted as a function of amino acid residue position in Figure 5D. The *ϕ* values show an increasing trend as the amino acid sites move from the surface towards the interior of the membrane and then decrease on the other side of the helix. This suggested the transmembrane domains (TMDs) of Q1-VSD span the width of the membrane bilayers. The negative value of *ϕ* indicated that the residue under study was solvent-exposed and hence not interacting with the membrane.
(3)ϕ=ln(ΔP1∕2(O2)ΔP12(NiEDDA)¯)
where Δ*P*_1/2_(*O*_2_) is the difference in *P*_1/2_ values of air and nitrogen exposed samples, and Δ*P*_1/2_*(NiEDDA)* is the difference in the *P*_1/2_ values for NiEDDA and nitrogen exposed samples. The *P*_1/2_ is the power where the first derivative amplitude is reduced to half of its unsaturated value.

Another recent example of using nitroxide spin labeling CW-EPR spectroscopy is the study of pinholin S^21^68 [66]. Pinholin S^21^68 is an essential part of the phage Φ21 lytic protein system that releases the virus progeny at the end of the infection cycle. TMD1 of active pinholin S^21^68 externalizes very quickly to the periplasm resulting in the active dimer. Within seconds of pinholin triggering the system, it forms heptametric holes by rapid oligomerization and reorientation of TMD2. Ahammad et al. analyzed CW-EPR spectra collected for spin-labeled active pinholin S^21^68 to investigate the dynamic properties of the active form of pinholin S^21^68 in 1,2-dimyristoyl-sn-glycero-3-phosphocholine (DMPC) lipid bilayers [66]. The CW-EPR spectral line shape analysis of the R1 side chain for 39 residue positions of S^21^68 suggested that the transmembrane domains (TMDs) have more restricted mobility when compared to the N- and C-termini. CW-EPR power saturation data collected on 31 spin-labeled sites of active pinholin S^21^68 in DMPC lipid bilayers suggested that the N-terminal remains in the periplasm and the TMD1 lies on the surface of the lipid bilayer with some residues pointing out of the lipid bilayer and others residues buried in the lipid environment. TMD2 remains incorporated in the lipid bilayer with the C-terminal of the S^21^68 in the cytoplasm. This study further predicted a tentative structural topology model of S^21^68 in lipid bilayers. Figure 6 shows the membrane depth parameter as a function of the active pinholin S^21^68 residue position in DMPC lipid-bilayered vesicles at room temperature and the proposed topology model of the S^21^68 in DMPC lipid bilayers [66].

SDSL CW-EPR spectroscopy at the X-band can also be used to study membrane topology of membrane proteins/peptides bound to aligned phospholipid bilayers [86,87,88,89,90]. 

### 4.2. Electron Spin Echo Envelope Modulation (ESEEM) Spectroscopy for Investigating the Local Secondary Structure of Protein/Peptides

ESEEM spectroscopy is a pulsed EPR technique that is sensitive to systems containing weak dipolar couplings between an electron spin and a NMR-active nuclear spin. It can provide great insight into the structure and function of many important biological systems [91,92,93,94,95,96,97,98,99,100,101,102,103,104,105,106]. This ESEEM technique can measure a distance between a spin label and a single ^2^H nucleus up to ~8 Å [107]. Nitroxide-based site-directed spin labeling ESEEM is very useful for probing the local secondary structure of membrane proteins/peptides in different environments including aqueous and lipid membranes [96,97,98,99,100,101,102,103,104,105]. The local secondary structure of membrane proteins has a great influence in the assembly, packing, and interaction of membrane proteins with their lipid membrane environment and hence is useful for understanding the function, dynamics, and interacting mode of membrane proteins [108,109]. 

In this ESEEM approach, a cysteine mutated nitroxide spin label is positioned 2 (*i+/−2*), 3 (*i+/−3*) or 4 (*i+/−4*) residues away from a fully deuterated valine or leucine side-chain (*i*). The characteristic periodicity of an α-helix (3.6 residue per turn with a pitch of 5.4 Å) structure gives rise to a unique pattern in the corresponding ESEEM spectra. At the X-band, a ^2^H ESEEM peak in the fourier transform (FT) frequency domain data is observed at ~2.2 MHz for the (*i+/−3*) or 4 (*i+/−4*) samples, whereas no ^2^H ESEEM peak is observed for the (*i+/−2*) sample or the control sample with no ^2^H [100]. For the (*i+/−2*) samples, spin labels are too far away from the ^2^H labeled valine/leucine to be detected. These unique patterns provide pertinent local secondary structural information on α-helical structural motifs for protein/peptides using this ESEEM spectroscopic approach with short data acquisition times (~30 min) and small sample concentrations (~100 µM). This ESSEM approach has been applied to several biologically important protein/peptide systems such as the acetylcholine receptor (AChR) M2δ peptide, ubiquitin peptide, amphipathic model peptide LRL_8_, intermediate filament protein human vimentin, and KCNE1 to probe their local secondary structures [96,97,98,99,100,101,102,103,104,105]. Figure 7 shows a recent example of the three-pulse ESEEM frequency domain data of KCNE1 ^2^H labeled Val in DMPC/DHPC bicelles [100]. The *i+3* and *i+4* samples show a peak at the ^2^H Larmor frequency at ~2.2 MHz. No peaks are observed for the control sample without ^2^H Val or the *i+2* sample.

### 4.3. SDSL EPR for Distance Measurement of Membrane Proteins

Double site-directed spin labeling of biological systems coupled with EPR spectroscopy is a very powerful and rapidly growing structural biology tool to measure distances between two spin labels for studying secondary, tertiary, and quaternary structures of macromolecules [20,50,110]. The magnetic dipolar interaction between two spin labels is inversely proportional to the cube of the distance (r^3^) and hence can be utilized for distance measurements [20]. This method can also be used to determine the relative orientations between interacting spin labels [111].

#### 4.3.1. CW-Dipolar Line Broadening SDSL EPR for Distance Measurement

Electron–electron dipolar interactions significantly broaden the CW-EPR spectral lineshape if the distance between the two unpaired electron spins is less than 20 Å. The strength of the dipolar interaction is estimated qualitatively from the degree of line broadening using a variety of lineshape analysis techniques to obtain distance information [110,111,112,113,114,115]. An intermediate distance range of 8–20 Å can be measured from the CW dipolar broadening EPR spectra and reveal important structural and dynamic information about membrane proteins [113]. SDSL CW dipolar broadening EPR has been applied to several important biological systems such as bacteriorhodopsin, sensory rhodopsin II (NpSRII)/transducer NpHtrII from natronobacterium pharaonis, erythroid β spectrin, AchR M2δ peptide, magainin 2 peptide, WALP peptide, bacterial K^+^-translocating protein KtrB, *E. coli* integral membrane sulfurtransferase (YgaP), proteorhodopsin oligomers, S-component ThiT from energy coupling factor (ECF) transporters, and KCNE1 [87,115,116,117,118,119,120,121,122,123,124,125,126,127].

#### 4.3.2. Double Electron Electron Resonance (DEER) Techniques for Distance Measurements 

DEER is also known as pulsed electron double resonance (PELDOR). DEER has been a widely used biophysical technique for measuring distances between two spin labels on membrane proteins in the range of 18–60 Å [128,129,130]. In DEER spectroscopy, a dipolar coupling between two spins is measured by monitoring one set of spins while exciting another set of spins with a second microwave frequency, leading to the measurement of the distance between them [36,128,131]. Nitroxide spin labeling based DEER spectroscopy is very popular for investigating the secondary, tertiary, and quaternary structures and conformational dynamics of a wide variety of macromolecules [19,44,57,61,120,121,122,123,124,125,126,127,128,129,130,132,133,134,135,136,137]. In addition to nitroxide spin labels, other spin labels such as functionalized chelators of paramagnetic lanthanides (Gd^III^), carbon-based radicals (trityl), and metals such as copper (Cu^II^) have been recently utilized for DEER measurements on membrane proteins [56,138,139,140,141]. There is also a disadvantage to using non-nitroxide spin labels. The Gd-based and trityl labels are bulkier than nitrixide spin labels, which can cause perturbation in protein structure and function [56]. Hence, care must be taken while choosing spin-labeling sites to avoid these perturbations. Figure 8 shows the DEER distance measurement method used for studying membrane proteins. The dipolar coupling frequency (ν_12_) is inversely related to the third power of the distance between two spin labels (ν_12_ ∝ *1/d*^3^) [36,142]. The most commonly used four pulse DEER sequence is shown in Figure 8B. In the four pulse DEER sequence, an echo is generated by applying three microwave pulses with specific positions to the one set of spins S_1_ at the probe frequency ν_1._ Another set of spins, S_2_, is flipped by applying a 4th pump microwave pulse at varying positions between the last two probe pulses at the frequency ν_2_. Consequently, the sign of the dipolar interaction and the amplitude of spin echo change result in the modulation of the echo amplitude as a function of the position of the pump pulse. The forward five-pulse and the reverse five-pulse DEER sequences are shown in Figure 8C,D, respectively. In the five-pulse DEER sequence (Figure 8C,D), similar lengths of the inter-pulse delays are applied. This minimizes the effect of the spin diffusion on relaxation, leading to the increase in the refocused echo intensity when compared to that of the four-pulse DEER sequence. The additional pump pulse also helps extend the dipolar evolution window [143,144,145]. A seven-pulse Carr-Purcell PELDOR sequence with multiple pump pulses (see Figure 8E) also leads to improved sensitivity in the measurement of long-range distances. [143,144,146]. These multipulse DEER experiments introduce echo crossing artifacts in DEER traces [143,144,146]. These artifacts can be minimized by using eight to thirty two-step phase cycling schemes [143].

##### Challenges and Methodological Development in DEER Measurements for Membrane Proteins

Despite the wide application of pulse DEER EPR techniques in structure biology, accurate and precise distance measurements are limited due to difficulties in integral membrane protein sample preparation in their functional environment. The heterogeneous distribution of spin-labeled proteins within the membrane creates local inhomogeneous pockets of high spin concentration leading to much shorter transverse relaxation/phase memory times and poor DEER modulation in more biologically relevant proteoliposomes when compared to water soluble proteins or membrane proteins in detergent micelles [44,147]. The proton spin diffusion further causes a decrease in the phase memory time. The proton spin diffusion arises due to the presence of hydrogens in the acyl chains of the lipid in addition to those in the solvent and in the protein [56]. The requirement of a high effective protein concentration in the liposome samples further introduces a strong background contribution that reduces sensitivity, distance range, and experimental throughput [148]. Additionally, the spin-labeled rotameric motions and protein backbone dynamics also contribute significantly to the width of the DEER distance distribution. 

Excellent work has been done in recent years to minimize these limitations. Sample preparations for the reconstitution of membrane proteins have been optimized in the presence of unlabeled proteins, bicelles, nanodics, lipodisq nanoparticles, a low protein/lipid molar ratio, and restricted spin label probes [44,136,147,149,150,151,152,153,154,155,156]. Using deuterated protein and solvents can also enhance the phase memory times that contribute towards the improvement of data quality [56]. DEER measurements are also conducted at Q-band to increase sensitivity [44,147,153]. The introduction of an arbitrary waveform generator (AWG) to EPR has opened new possibilities to improve the pulse sequences in DEER experiments [143]. The increased excitation band width of the linear chirp pump pulses enhances the modulation depth. This increases the sensitivity of the DEER experiment [56,143,146]. The use of 5-pulse and 7-pulse sequences also help to increase the dipolar evolution window [56,143]. Computational approaches using molecular dynamics simulations using DEER distance restraints have also been widely used to refine the structural properties of membrane proteins [60,136,157,158,159]. These methodological developments have made nitroxide based DEER spectroscopy a rapidly expanding structural biology tool to study complicated integral membrane protein systems. 

Nitroxide based SDSL DEER spectroscopy has been applied to investigate a variety of membrane protein systems such as *E. coli* integral membrane sulfurtransferase (YgaP), pentameric ligand-gated channel, homodimer protein, bacteriorhodopsin, KCNE1, KCNE3, C99 amyloid precursor protein, KvAP voltage-sensing domain, human dihydroorotate dehydrogenase enzyme (HsDHODH), influenza A M2 protein, outer membrane cobalamin transporter BtuB in intact *E. coli*, cardiac Na^+^/Ca^2+^ exchange (NCX1.1) protein, Na^+^/Proline transporter PutP *Escherichia coli*, tetrameric potassium ion channel KcsA, α-synuclein, membrane-fusion K/E peptides, ABC transporter MsbA, HCN channels, YetJ membrane protein, ectodomain of gp41, and multidrug transporter LmrP [60,118,136,149,150,151,155,157,160,161,162,163,164,165,166,167,168,169,170,171,172,173]. SDSL DEER spectroscopy has been recently used to study the oligomerization states of several membrane proteins such as NhaA Na^+^/H^+^ antiporter of E-coli, KcsA, M2 transmembrane domain, LptA, proteorhodopsin, and Bax oligomers [64,134,158,161,174,175,176,177,178,179]. 

A recent application of nitroxide-based spin labeling DEER spectroscopy is the study of a YetJ membrane protein [155]. YetJ is a member of the widely distributed transmembrane Bax inhibitor motive (TMBIM) family found to be important for the uptake of calcium into bacteria and in mediating a pH-dependent Ca^2+^ flux in proteoliposomes. YetJ has seven transmembrane helices with 214 amino acid residues. Li et al. carried out DEER distance measurements on the dual spin-labeled YetJ mutant 44R1/152R1 reconstituted into nanodiscs (NDs) [155]. This experiment revealed a bimodel-like distance distribution displaying two major peaks centered at 2.7 and 4.1 nm suggesting two conformations of YetJ in NDs [155]. This study further revealed that the use of the nanodiscs (NDs) provided improvement in the overall signal-to-noise ratio (S/R) of DEER signals and hence increased the resolution in the distance distribution [155]. 

DEER spectroscopy was recently applied to study conformational changes in the extracellular loops of the outer membrane cobalamin transporter BtuB in intact *E. coli* [179]. BtuB is a 22-stranded β-barrel protein consisting of a 130 residue N-terminal plug or hatch domain in the center. It is a member of the TonB-dependent transporter (TBDT) family. It requires a proton motive force (pmf) and the inner membrane ExbB-ExbD-TonB complex for cyanocobalamin (CN-Cbl) transport. Josesh et al. performed DEER distance measurements on the 188R1-399R1 mutant in *E. coli* cells [179]. DEER distance measurements on 188R1-399R1 cells showed a shorter distance in the apo-state (no Ca^2+^ or CN-Cbl) in *E. coli* when compared to that in the presence of Ca^2+^ or CN-Cbl, suggesting conformation changes induced by ligand binding [179].

Another recent example of using SDSL DEER spectroscopy is the study of multimeric membrane transport proteins such as sodium and aspartate symporter from *Pyrococcus horikoshii*, Glt_ph_ [180,181]. Glt_ph_ is a homologue of the mammalian glutamate transporters. It is a homo-trimeric integral membrane protein that controls the neurotransmitter levels in brain synapses [181]. Glt_Ph_ transports aspartate together with three Na^+^ ions into the cytoplasm accompanied by stoichiometrically uncoupled Cl^−^ conductance [180]. Each subunit of Glt_ph_ has a complex topology with eight transmembrane and two reentrant hairpin segments. Georgieva et al. carried out extensive DEER distance measurements on eight transport domain mutants in detergent and lipid membranes either in the apo state or bound to Na^+^ and asp or to DL-*threo*-β-bezyloxyaspartate (TBOA) [181]. These results suggested that the GltPh protomers are distributed between the outward and inward facing conformations in detergent solutions and in lipid bilayers both in the bound and unbound states [181]. Riederer et al. further utilized DEER distance measurements to evaluate the intra-subunit structural changes in Glt_ph_ [180]. The DEER measurements on the homomeric V216C/I294C mutants in the presence of Na^+^ ions and Asp revealed broad, weakly-structured distance distributions, spanning from 40 to 70 Å, whereas, the heteromeric protein exhibited a single narrow peak at 52 Å [180]. These distance distributions suggested that the V216C/I294C mutant subunit populates the inward facing state. The corresponding distance in the outward facing state is 34 Å [180]. These results allowed for straightforward determination of the conformational state of the labeled promoter. 

Another recent example of DEER spectroscopy in the literature is an investigation into the active oligomeric Bax proteins [177,182]. Bax is a Bcl-2 protein that plays a crucial role in apoptosis regulation and execution. Cytosolic Bax monomers oligomerize, when activated, on the surface of mitochondria and change their conformation to form holes into the outer membrane. These proteins are interesting targets for drug development for treatment of chronic lymphocytic leukemia. Bleicken et al. utilized SDSL DEER spectroscopic measurements to obtain 42 distances between 12 different spin-labeled positions of full-length Bax in lipid bilayers mimicking the mitochondrial outer membrane (MOM) [182]. These distance constraints were used to calculate a three-dimensional structural model of full length active Bax in the membrane. These results suggested that active Bax is organized as assemblies of dimers at the membrane. Additionally, each monomer contains a more flexible domain involved in interdimer interactions and pore formation. Teucher et al. further utilized a combination of spectroscopically distinguishable nitroxide and gadolinium spin labels for DEER measurements to investigate the quaternary structure of active and membrane-embedded Bax oligomers [177]. DEER distance measurements on the mixture of the Bax_C87Gd_ with Bax_WTR1_ forming a complex protein homo-oligomer detected NO-NO, NO-Gd, and Gd-Gd distances independently. When Bax_WTR1_ was mixed with a three-fold excess of Bax_C87Gd_, a Gd-Gd distance of 6 nm suggested interactions between the two Bax_C87Gd_ monomers within a dimer unit, and NO-Gd distances in the range of 2.5–5 nm confirmed that the Bax oligomers containing Bax_C87Gd_ and Bax_WTR1_ monomers are formed [177]. Figure 9 shows DEER distance measurements on the Bax oligomers [177]. The background was subtracted from the raw DEER time domain data as shown by the dotted lines in the left panel and the distance distribution (right panel) was obtained using Tikhonov regularization with DeerAnalysis2018 [177]. This study also highlights the usefulness of the orthogonal spin-labeling-based DEER spectroscopy to monitor the membrane-embedded homo- and hetero-oligomers of Bcl-2 proteins. 

## 5. Conclusions

In this review, we discussed some recent applications of nitroxide-based SDSL EPR spectroscopic techniques to study biologically important membrane protein systems. With recent methodological developments, SDSL EPR spectroscopy has become a very popular and rapidly growing structure biology technique used to answer pertinent structural and dynamic questions related to biological systems. It can provide important information on complicated biological systems that is very difficult or impossible to discover by using other conventional biophysical techniques. 

## Figures and Tables

**Figure 1 biomolecules-10-00763-f001:**
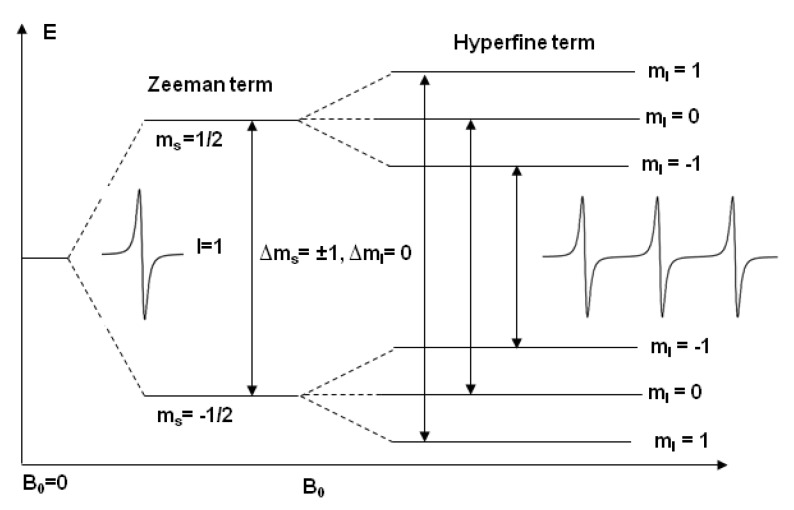
Energy level diagram of a nitroxide spin label in the presence of a static magnetic field (B_0_). The inset traces show the corresponding electron paramagnetic resonance (EPR) spectrum in the absence and presence of a ^14^N (I = 1) hyperfine interaction [19].

**Figure 2 biomolecules-10-00763-f002:**
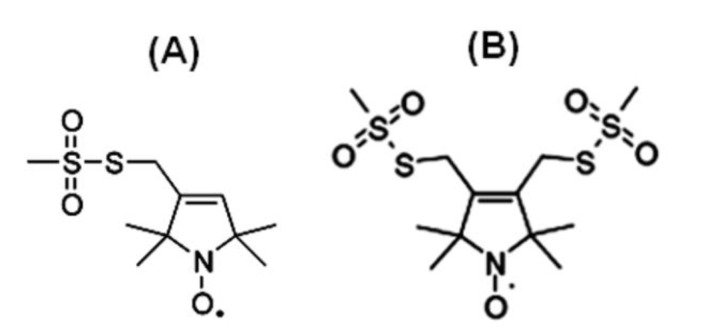
Structure of examples of nitroxide spin labels used in the site-directed spin labeling (SDSL) EPR study of membrane proteins. (**A**) Methanethiosulfonate spin label (MTSL), (**B**) Bifunctional spin label (BSL).

**Figure 3 biomolecules-10-00763-f003:**
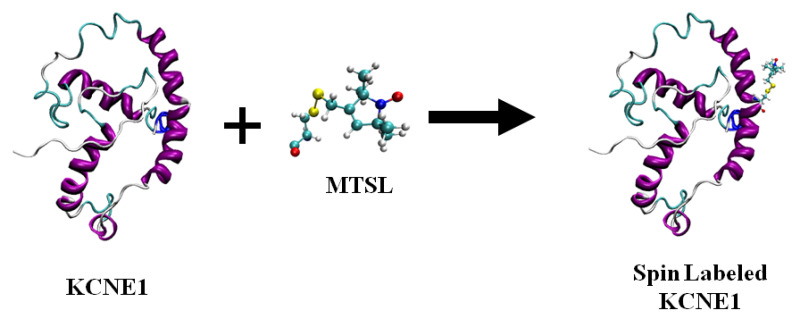
A cartoon representation of the structure of MTSL (methanethiosulfonate spin label) and the resulting side-chain produced by reaction with a cysteine residue (T58C) on a KCNE1 membrane protein. The cartoon structure of the MTSL-labeled KCNE1 (PDB ID:2k21) was rendered using visual molecular dynamics (VMD) [52,53].

**Figure 4 biomolecules-10-00763-f004:**
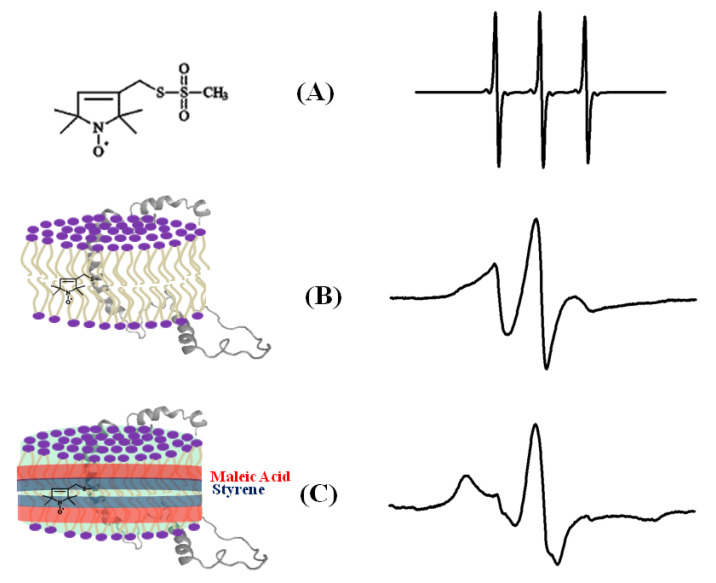
An illustrative example of the EPR spectra for a spin-labeled membrane protein in different membrane environments. (**A**) A free MTSL spin label in solution, (**B**) MTSL spin label on a F56 C-KCNE1 membrane protein in a lipid bilayer, (**C**) MTSL spin label on a F56C-KCNE1 membrane protein in lipodisq nanoparticles. The CW-EPR spectrum for lipodisq nanoparticle samples also shows a minor peak due to free spin labels. (Adapted from [69] with permission).

**Figure 5 biomolecules-10-00763-f005:**
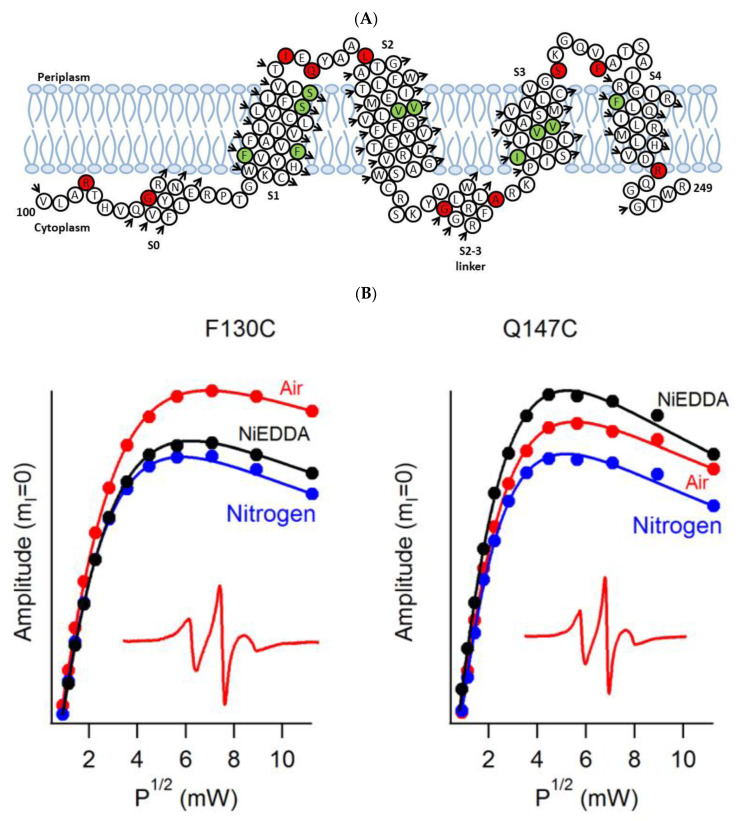
(**A**) The proposed topology of the Q1VSD sequence in lipid bilayers. The black arrows show the order of the amino acid residues in the protein sequence. The green and red circles represent spin-label sites buried inside and outside of the membrane bilayers respectively. (**B**) and (**C**) EPR power saturation curves from Q1VSD in 1-palmitoyl-2-oleoyl-phosphatidylcholine (POPC)/1-palmitoyl-2-oleoyl-phosphatidylglycerol (POPG) lipid-bilayered vesicles at 295 K. Mutation F130C is on helix S1 and is a part of the transmembrane domain, while the Q147C site is at the linker between helix S1 and helix S2 at a site outside the lipid bilayer. Mutation F232C is on helix S4 and is a part of the transmembrane domain, while the F222C site is at the linker between helix S3 and helix S4 at a site outside the lipid bilayer. The inset spectra are the corresponding CW-EPR spectra for these sites. (**D**) Membrane depth parameter (*ϕ*) as a function of Q1VSD residue position in POPC/POPG lipid-bilayered vesicles at 295 K. (Adapted from [84] with permission).

**Figure 6 biomolecules-10-00763-f006:**
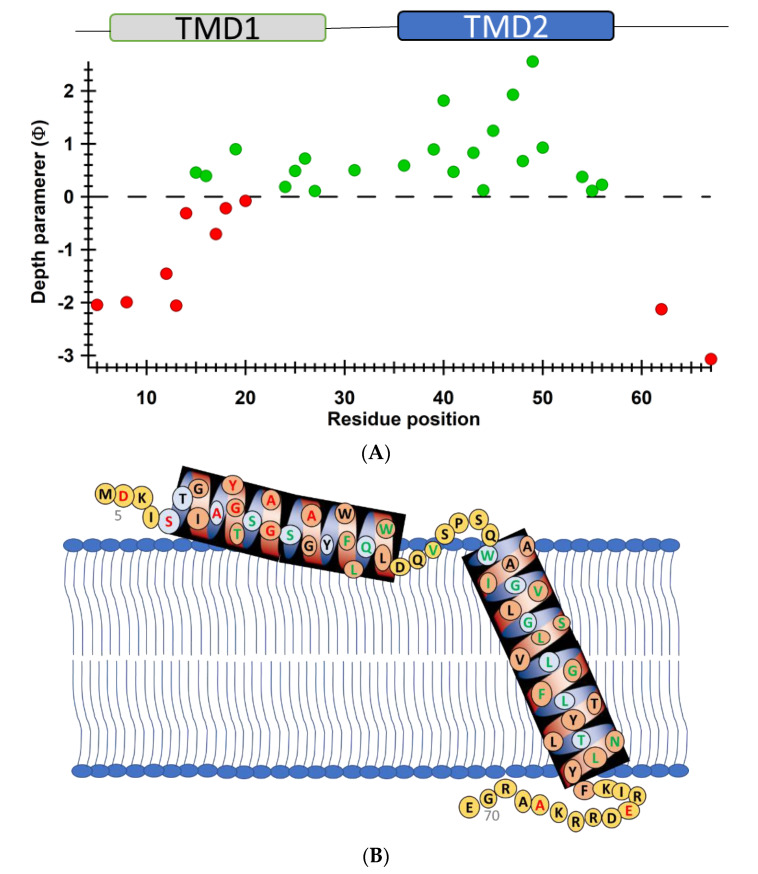
(**A**) Membrane depth parameter (*ϕ*) as a function of the S^21^68 residue positions in DMPC lipid-bilayered vesicles at room temperature. (**B**) The proposed topology model of the S^21^68 in DMPC lipid bilayers. (Adapted from [66] with permission).

**Figure 7 biomolecules-10-00763-f007:**
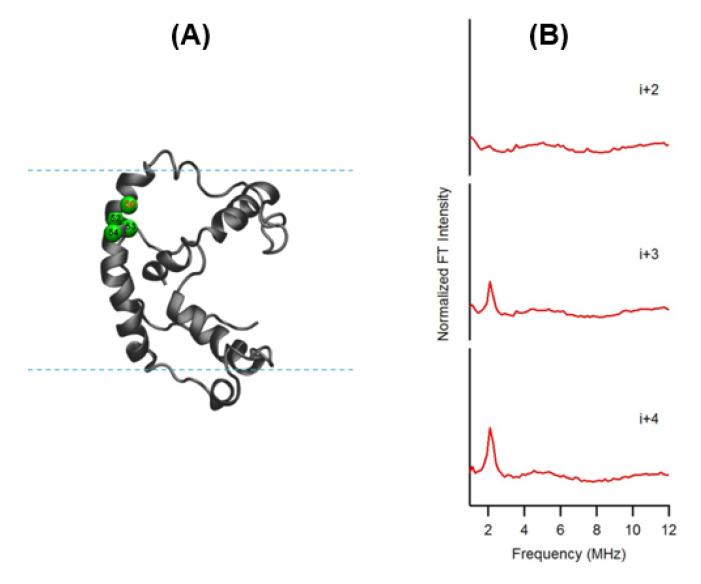
(**A**) Cartoon representation of KCNE1 in DMPC/ dodecylphosphocholine (DPC) bicelles. The probed α-helical region is colored in green and located on the transmembrane domain of the full-length KCNE1. Residue 50 is side-chain ^2^H-labeled Val (denoted i), Residues 52, 53, and 54 are independent Cys mutations (denoted *i+2*, *i+3*, and *i+4*, respectively). (**B**) Frequency domain spectra of three-pulse ESEEM data of *i+2*, *i+3*, and *i+4* samples shown in normalized FT intensity. (Adapted from [100] with permission).

**Figure 8 biomolecules-10-00763-f008:**
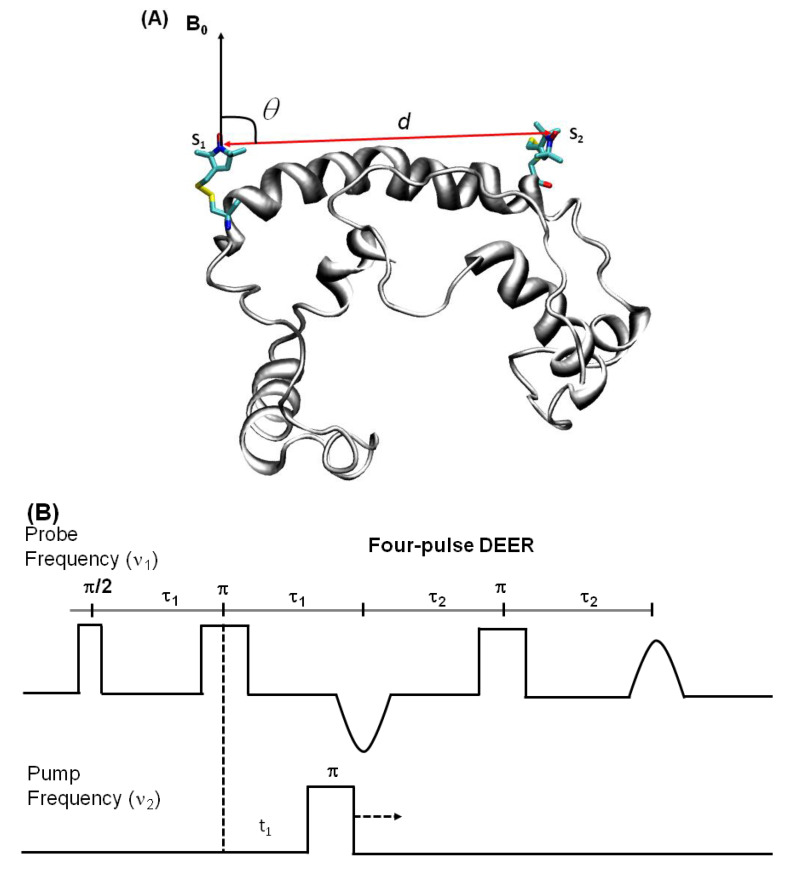
Double electron electron resonance DEER spectroscopic method used to measure distances between nitroxide spin labels. (**A**) Distance vector (d) between spin S_1_ and S_2_ on KCNE1 membrane protein (PDB ID: 2k21) at an angle θ with the magnetic field B_0_. (**B**) Four-pulse DEER sequence. (**C**) Forward five-pulse DEER sequence. (**D**) Reverse five-pulse DEER sequence. (**E**) Seven-pulse CP-PELDOR sequence [143,144].

**Figure 9 biomolecules-10-00763-f009:**
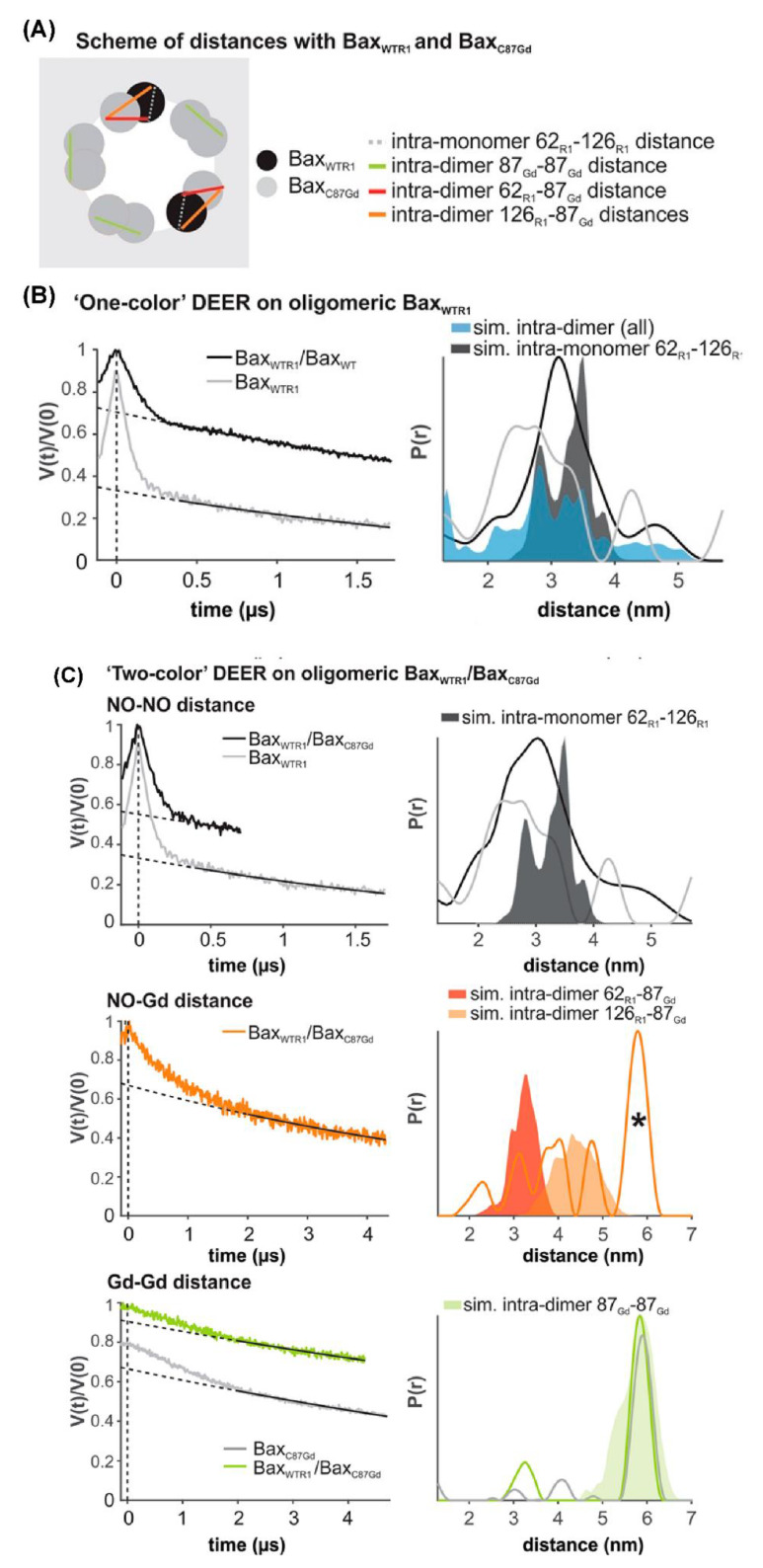
DEER distance measurements on oligomeric Bax with orthogonal spin labels. (**A**) Schematic of the expected distances. (**B**,**C**) Primary data with background function (left) and time distribution (right). The shaded areas represent corresponding distance simulations based on the structures. (**B**) Active Bax_WTR1_ mixed with 3-fold excess unlabeled Bax_WT_ (black) compared to the undiluted Bax_WTR1_ (grey). (**C**) Upper panel, NO-NO DEER on active Bax_WT_ with 3-fold excess of Bax_C87Gd_ compared to the Bax_WT_ alone (grey). Central panel, NO-Gd DEER on the samples. The asterisk highlights a possible channel cross-talk signal. Bottom panel, Gd-Gd DEER on the sample (green) compared to Bax_C87Gd_ alone (Grey). (Adapted from [177] with permission).

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
