# Peer review of "Electron Paramagnetic Resonance as a Tool for Studying Membrane Proteins"

_biomolecules, 2020, doi:10.3390/biom10050763_

Round 1

Reviewer 1 Report

This review gives an extensive overview of the different EPR spectroscopy methodologies used for studying membrane proteins. In general the review is written well and covers the different methods typically used in EPR spectroscopy of membrane proteins. However. in my opinion this review could be improved by providing a bit more technical information regarding data analysis methods and I will give some examples:

a) In section 4.1 the authors discuss the use of CW-EPR for studying dynamics giving an example in figure 4 demonstrating broadening of the line as a result of motional constraints. this discussion is missing in my opinion the option to obtain quantitative information about the motion correlation times using freely available simulation packages such as easyspin. Additionally I think a discussion of the analysis of this type of spectra cannot be complete without quoting the work of Freed who laid the foundations for analyzing these spectra for example in:  

Freed JHTheory of slow tumbling ESR spectra of nitroxides. In: LJ Berliner, ed. Spin Labeling: Theory and ApplicationsNew York: Academic Press; 197653‐ 132

b) Figure 5 shows an example of the use of power saturation experiments for studyine the topolgy of the protein with respect to the membrane. Here again I think it is worthwhile to give some description of how the data analysis works and what is the quantitative information extracted and hoe this is done. The figure includes data and a fit of the data without any explanation about this fit.

c) On the same line Figure 8 shows DEER raw data and distance distributions extracted. I would expect to at least explain that the DEER data includes a background component described by the dashed line in the figure.

I am aware that this journal is more biologically oriented and the review is targeting this type of readers but I am still in the opinion that some more technical information would help these readers understand the complexity of the data analysis and the assumptions being made in the data analysis.

Author Response

This review gives an extensive overview of the different EPR spectroscopy methodologies used for studying membrane proteins. In general the review is written well and covers the different methods typically used in EPR spectroscopy of membrane proteins. However. in my opinion this review could be improved by providing a bit more technical information regarding data analysis methods and I will give some examples:

a) In section 4.1 the authors discuss the use of CW-EPR for studying dynamics giving an example in figure 4 demonstrating broadening of the line as a result of motional constraints. this discussion is missing in my opinion the option to obtain quantitative information about the motion correlation times using freely available simulation packages such as easyspin. Additionally I think a discussion of the analysis of this type of spectra cannot be complete without quoting the work of Freed who laid the foundations for analyzing these spectra for example in:  

Freed JH. Theory of slow tumbling ESR spectra of nitroxides. In: LJ Berliner, ed. Spin Labeling: Theory and Applications. New York: Academic Press; 1976: 53‐ 132. 

As suggested, the following description has been included in the revised manuscript (see page 5)

"

 The more quantitative information about the spin label side-chain dynamics can be obtained with EPR spectral simulation approaches using freely available simulation programs such as  Easyspin, and Non- linear least squares (NLSL) [59, 67, 68].

b) Figure 5 shows an example of the use of power saturation experiments for studyine the topolgy of the protein with respect to the membrane. Here again I think it is worthwhile to give some description of how the data analysis works and what is the quantitative information extracted and hoe this is done. The figure includes data and a fit of the data without any explanation about this fit.

As suggested, the following description has been included in the revised manuscript (see page 6):

" The CW-EPR power saturation data were analyzed to obtain peak-to-peak amplitude of the first derivative mI =0 resonance line and plotted against the square root of the incident microwave power for three sample conditions: 1) equilibrated with nitrogen as a control; 2) equilibrated with lipid-soluble paramagnetic reagent 20% oxygen (air); and 3) equilibrated with nitrogen in the presence of a water-soluble paramagnetic reagent NiEDDA. Figures 5B and 5C show that residues Q147C and F222C of Q1-VSD have greater accessibility to NiEDDA, while F130C and F232C appear to interact more with O2 in the non-polar lipid environment. The membrane depth parameter (f) obtained from CW-EPR power saturation data (using equation 3) were plotted as a function of amino acid residue position in Figure 5D. The f values show an increasing trend as the amino acid sites move from the surface towards the interior of the membrane and then decrease on the other side of the helix. This suggested the transmembrane domains (TMDs)  of Q1-VSD span the width of the membrane bilayers. The negative value of f indicated that the residue under study was solvent exposed and hence not interacting with the membrane.

                                            ,                  (3)

where ∆P1/2(O2) is the difference in P1/2 values of air and nitrogen exposed samples, and ∆P1/2(NiEDDA) is the difference in the P1/2 values for NiEDDA and nitrogen exposed samples. The P1/2 is the power where the first derivative amplitude is reduced to half of its unsaturated value.

c) On the same line Figure 8 shows DEER raw data and distance distributions extracted. I would expect to at least explain that the DEER data includes a background component described by the dashed line in the figure.

As suggested, the following description has been included in the revised manuscript (see page 14)

"

The background was subtracted from the raw DEER time domain data as shown by the dotted lines in the left panel and the distance distribution (right panel) was obtained using Tikhonov regularization with DeerAnalysis2018 [172]. "

I am aware that this journal is more biologically oriented and the review is targeting this type of readers but I am still in the opinion that some more technical information would help these readers understand the complexity of the data analysis and the assumptions being made in the data analysis.

We have updated the manuscript with additional technical information as suggested by the reviewer.

Reviewer 2 Report

The manuscript „Electron Paramagnetic Resonance as a Tool for Studying Membrane Proteins“ by Sahu & Lorigan is a brief review of EPR spectroscopy as a tool to study membrane proteins.

The topic of the review is interesting but there are some major flaws that need to be addressed before the manuscript can be accepted for publication. In addition to the major problems listed below, I find that the review is not very well written. There are many orthographic problems and several sentences have missing words. Some examples are given below.

Major:

  • In my opinion, the figures are not very well described and connected to the main text:
  1. Figure 2 is referenced in the text as showing the most widely used nitroxide spin labels. I do not think that this is true for the Rx label.
  2. In Figure 3, the protein looks completely different after spin labelling. This is because the structure is turned upside down. It gives the impression that the structure changes due to the spin label. Also, the spin label is a bit hard to see in the labelled structure, since it is not shown as ball and stick as on the educt side of the reaction.
  3. In Figure 4, the spin labels are almost invisible in the B) and C) panels. Also, it might be worth mentioning in the caption that some free label is visible in the cw spectrum in panel C). Which residue was labelled?
  4. Figure 5A): The meaning of the black arrows is not explained anywhere. Also, what is the meaning of the red and green residues? I suspect they correspond to panel D), but that should really be described in the caption.
  5. Figure 5B-D) It is not explained (also not in the main text) how these graphs have to be interpreted. How is the depth parameter derived?
  6. Figure 8) The resolution is very low and it is hard to read the subscripts, which makes it difficult to compare the figure with the main text.
  • The different EPR methods are explained extremely briefly. It is my impression that the review is targeted for non EPR experts who are interested to learn the possibilities that the method offers. Therefore, I think that for example a schematic about how DEER/PELDOR works would be much more helpful than Figure 1, for example. Also, the NiEDDA methodoly is not explained, although results from this method are shown in a page-filling figure.

Minor (some examples):

Page 1, line 28: … are known as…

Page 1, line 34: many instead of several ?

Page 1, line 35: The sentence does not make sense.

Page 1, lines 34-43: Many sentences begin with “Membrane proteins”

Page 2, line 49: There are many instances where X-ray structures in different conformations have been solved. This provides dynamic information. Without such structures, it would be very difficult to make sense of many other experiments (e.g. DEER data)

Page 2, line 64: How is this different from EPR?

Page 2, line 70: “… to the NMR.” Sounds a bit strange.

Page 2, line 72-73: “… with certain nuclei of the individual atom … ” is this correct?

Page 3, line 94: irradiation?

Page 3, line 112: It should be mentioned that DEER is also known as PELDOR. This has been done further below, but I think it should be mentioned right at the top.

Page 4, line 134: Rx is not the most widely used label

Page 4, line 138: “… the cysteine residue…” does KCNE1 have just one? Is it artificial? What is the residue number?

Page 4, line 142: rendered instead of obtained

Page 5, line 151: “… provides its motion highly dependent…”? Does not make sense.

Page 5, line 156: If it moves very slow, it is not motionless.

Page 11, line 330: E. coli should be in italics.

Author Response

The manuscript „Electron Paramagnetic Resonance as a Tool for Studying Membrane Proteins“ by Sahu & Lorigan is a brief review of EPR spectroscopy as a tool to study membrane proteins.

The topic of the review is interesting but there are some major flaws that need to be addressed before the manuscript can be accepted for publication. In addition to the major problems listed below, I find that the review is not very well written. There are many orthographic problems and several sentences have missing words. Some examples are given below.

Major:

  • In my opinion, the figures are not very well described and connected to the main text:

1. Figure 2 is referenced in the text as showing the most widely used nitroxide spin labels. I do not think that this is true for the Rx label.

As suggested, we have updated the revised manuscript reflecting the reviewer's concern (see page 4):

2. In Figure 3, the protein looks completely different after spin labelling. This is because the structure is turned upside down. It gives the impression that the structure changes due to the spin label. Also, the spin label is a bit hard to see in the labelled structure, since it is not shown as ball and stick as on the educt side of the reaction.

We have updated the figure as suggested by the reviewer. We also added the ball and stick representation of the spin label (see Figure 3).

3. In Figure 4, the spin labels are almost invisible in the B) and C) panels. Also, it might be worth mentioning in the caption that some free label is visible in the cw spectrum in panel C). Which residue was labelled?

As suggested, we have modified the figure with higher resolution of the spin label and the caption has been also updated with the spin label position (see Figure 4 and its caption).

4. Figure 5A): The meaning of the black arrows is not explained anywhere. Also, what is the meaning of the red and green residues? I suspect they correspond to panel D), but that should really be described in the caption.

As suggested, we have updated the Figure caption as follows to address the reviewer's concern (see page 6, Figure 5 caption).

" The black arrows shows the order of the amino acid residue in the protein sequence. The green and red circles represent spin label sites buried inside and outside of the membrane bilayers respectively. "

5. Figure 5B-D) It is not explained (also not in the main text) how these graphs have to be interpreted. How is the depth parameter derived?

As suggested , we have included the following description in the revised manuscript to address the reviewer's concern (see page 6).

"

The CW-EPR power saturation data were analyzed to obtain peak-to-peak amplitude of the first derivative mI =0 resonance line and plotted against the square root of the incident microwave power for three sample conditions: 1) equilibrated with nitrogen as a control; 2) equilibrated with lipid-soluble paramagnetic reagent 20% oxygen (air); and 3) equilibrated with nitrogen in the presence of a water-soluble paramagnetic reagent NiEDDA. Figures 5B and 5C show that residues Q147C and F222C of Q1-VSD have greater accessibility to NiEDDA, while F130C and F232C appear to interact more with O2 in the non-polar lipid environment. The membrane depth parameter (f) obtained from CW-EPR power saturation data (using equation 3) were plotted as a function of amino acid residue position in Figure 5D. The f values show an increasing trend as the amino acid sites move from the surface towards the interior of the membrane and then decrease on the other side of the helix. This suggested the transmembrane domains (TMDs)  of Q1-VSD span the width of the membrane bilayers. The negative value of f indicated that the residue under study was solvent exposed and hence not interacting with the membrane.

                                            ,                  (3)

where ∆P1/2(O2) is the difference in P1/2 values of air and nitrogen exposed samples, and ∆P1/2(NiEDDA) is the difference in the P1/2 values for NiEDDA and nitrogen exposed samples. The P1/2 is the power where the first derivative amplitude is reduced to half of its unsaturated value.

"

6. Figure 8) The resolution is very low and it is hard to read the subscripts, which makes it difficult to compare the figure with the main text.

We have updated the figure with improved resolution to address the reviewer's concern. (see Figure 9).

  • The different EPR methods are explained extremely briefly. It is my impression that the review is targeted for non EPR experts who are interested to learn the possibilities that the method offers. Therefore, I think that for example a schematic about how DEER/PELDOR works would be much more helpful than Figure 1, for example.

As suggested by reviewer, we have included additional figure and explanation about  how DEER works in the revised manuscript (see page 11, 12 and Figure 8).

  • Also, the NiEDDA methodoly is not explained, although results from this method are shown in a page-filling figure.

As suggested, the NiEDDA methodology has been explained in the revised manuscript (see page 6) .

Minor (some examples):

Page 1, line 28: … are known as…

Corrected in the revised manuscript.

Page 1, line 34: many instead of several ?

As suggested, the revised manuscript has been updated.

Page 1, line 35: The sentence does not make sense.

We have modified the sentence as follows:

" The 30% of human genomes are known to be encoded by membrane proteins [2-4]."

Page 1, lines 34-43: Many sentences begin with “Membrane proteins”

We have updated the revised manuscript to address the reviewer's concern (see page 1).

Page 2, line 49: There are many instances where X-ray structures in different conformations have been solved. This provides dynamic information. Without such structures, it would be very difficult to make sense of many other experiments (e.g. DEER data)

That sentence was modified to reflect the reviewer's concern (see page2).

Page 2, line 64: How is this different from EPR?

The probe used in FRET is larger when compared to the spin label probes used in EPR. The incorporation of the probe is very challenging in FRET when compared to that of the spin label probe in EPR.

Page 2, line 70: “… to the NMR.” Sounds a bit strange.

The following changes have been made to address the reviewer's concern (see page 2).

" The working principle of the EPR is similar to the working principle of NMR. "

Page 2, line 72-73: “… with certain nuclei of the individual atom … ” is this correct?

The "certain nuclei" was replaced by "isotopic nuclei" in the revised manuscript. (see page 2)

Page 3, line 94: irradiation?

"irradiation" was removed in the revised manuscript.

Page 3, line 112: It should be mentioned that DEER is also known as PELDOR. This has been done further below, but I think it should be mentioned right at the top.

 As suggested, the that sentence was moved at the top in the revised manuscript. (see page 11)

Page 4, line 134: Rx is not the most widely used label

We have removed " the most widely used" in the revised manuscript. (see page 4)

Page 4, line 138: “… the cysteine residue…” does KCNE1 have just one? Is it artificial? What is the residue number?

As suggested we have replaced " cysteine residue " with " cysteine residue (T58C)" in the revised manuscript. (see page 4).

Page 4, line 142: rendered instead of obtained

updated in the revised manuscript.

Page 5, line 151: “… provides its motion highly dependent…”? Does not make sense.

We have updated the sentence as follows in the revised manuscript: (see page 5)

"

The flexibility of the MTSL nitroxide spin label provides its motion which is highly dependent on neighboring amino acid side chains and secondary structure components in its immediate environment."

Page 5, line 156: If it moves very slow, it is not motionless.

As suggested, we have changed "motionless" to " close to motionless" in the revised manuscript. (see page 5)

Page 11, line 330: E. coli should be in italics.

updated in the revised manuscript.

Reviewer 3 Report

Unfortunately, the present review does not add anything new to the field. The authors themselves had published a similar review in 2018 (https://doi.org/10.1155/2018/3248289). In addition, another review on the topic (from the group of E. Bordignon) with a more detailed discussion of the technique and applications had been published in the same year (https://doi.org/10.1016/j.bbamem.2017.12.009). The review could have been written in a less superficial manner. Also, the authors missed to include some of the state-of-the-art applications as well as to define the present challenges/solutions from a methodological and mechanistic perspective (in addition to a few the authors mentioned).

Author Response

Unfortunately, the present review does not add anything new to the field. The authors themselves had published a similar review in 2018 (https://doi.org/10.1155/2018/3248289). In addition, another review on the topic (from the group of E. Bordignon) with a more detailed discussion of the technique and applications had been published in the same year (https://doi.org/10.1016/j.bbamem.2017.12.009). The review could have been written in a less superficial manner. Also, the authors missed to include some of the state-of-the-art applications as well as to define the present challenges/solutions from a methodological and mechanistic perspective (in addition to a few the authors mentioned).

We have extensively revised the manuscript with the additional technical details and additional figures to improve the quality of the manuscript to address the reviewer's concern.

Round 2

Reviewer 2 Report

Most of my concerns regarding the figures have been addressed.

There are still many sloppy mistakes like missing labels in the figures or grammar mistakes.

This needs to be improved before the manuscript can be accepted.

Author Response

Most of my concerns regarding the figures have been addressed.

There are still many sloppy mistakes like missing labels in the figures or grammar mistakes.

This needs to be improved before the manuscript can be accepted.

As suggested by reviewer, we have revised the manuscript to improve the overall quality of the manuscript.

Reviewer 3 Report

The authors have improved the overall quality of the manuscript by adding further technical descriptions. However, other than describing a few of the important applications, I am not convinced that either the review adds anything new to the field or gives a comprehensive description of all the current developments. Some of major problems are listed below.

The description of the labels for membrane protein structural biology is incomplete. Other than nitroxide, Gd(III), Mn(II), and Trityl labels have been used with membrane proteins. Thus, the spin labeling part must be substantially improved. Although authors improved the DEER part, it is still lacking important details (of theoretical description and data analysis).  Here the authors need to include the 5-pulse / 7-pulse DEER and their applications for membrane proteins.  The new developments of DEER in native environments for membrane proteins are missing. Instead of filling the last two pages with figures from a previous publication, the authors could have optimized their size and include additional relevant figures. Moreover, the challenges and the methodological development part needs to be substantially improved.

Author Response

The description of the labels for membrane protein structural biology is incomplete. Other than nitroxide, Gd(III), Mn(II), and Trityl labels have been used with membrane proteins. Thus, the spin labeling part must be substantially improved.

As suggested, we have included the following description in the revised manuscript: (see page 11)

"In addition to nitroxide spin labels, other spin labels such as functionalized chelators of paramagnetic lanthanides (GdIII), carbon-based radicals ((trityl), metals such as copper (CuII) have been recently utilized for DEER measurements on membrane proteins [56, 139-142]."

Although authors improved the DEER part, it is still lacking important details (of theoretical description and data analysis).  Here the authors need to include the 5-pulse / 7-pulse DEER and their applications for membrane proteins.   The new developments of DEER in native environments for membrane proteins are missing. Instead of filling the last two pages with figures from a previous publication, the authors could have optimized their size and include additional relevant figures.

As suggested, we included 5-pulse and 7-pulse DEER sequences and corresponding description in the revised manuscript; (see pages 11, 12, 13, Figure 8C, D, E ). We have also included the additional applications of DEER spectroscopy in the revised manuscript to address the reviewer's concern.(see pages 14, 15).

Page, 11,

"In the five-pulse DEER sequence (Figure 8C, D), similar lengths of the inter-pulse delays are applied. This minimizes the effect of the spin diffusion on relaxation leading to the increase in the refocused echo intensity when compared to that of four-pulse DEER sequence. The additional pump pulse also helps extend the dipolar evolution window [144, 146]. A seven-pulse Carr-Purcell DEER sequence with multiple pump pulses (see Figure 8E) allows for more substantial increase in the length of the dipolar evolution time [144, 145, 147]."

Pages 14, 15,

"DEER spectroscopy was recently applied to study conformational changes in the extracellular loops of the outer membrane cobalamin transporter BtuB in intact E. coli [182].  BtuB is a 22-stranded β-barrel protein consisting of a 130 residue N-terminal plug or hatch domain in the center. It is a member of the TonB-dependent transporter (TBDT) family. It requires a proton motive force (pmf) and the inner membrane ExbB-ExbD-TonB complex for cyanocobalamin (CN-Cbl) transport. Josesh et al. performed DEER distance measurements on 188R1-399R1 mutant in E. coli cells [182]. DEER distance measurements on 188R1-399R1 cells showed a shorter distance in the apo-state (no Ca2+ or CN-Cbl) in E. coli when compared to that in the presence of Ca2+ or CN-Cbl suggesting a conformation changes induced by the ligand binding [182]."

Moreover, the challenges and the methodological development part needs to be substantially improved.

 As suggested, the following description has been included to improve the methodological development part. (see pages 13, 14)

"The proton spin diffusion further causes decrease in the phase memory time. The proton spin diffusion arises due to the presence of hydrogens in the acyl chains of the lipid in addition to those in the solvent and in the protein [56]."

"Using deuterated protein and solvents can also enhance the phase memory times that contributes towards the improvement of data quality[56]. DEER measurements are also conducted at Q-band to increase sensitivity [44, 148, 154]. The introduction of an arbitrary waveform generators (AWG) to EPR has opened new possibilities to improve the pulse sequences in DEER experiment144. The increased excitation band width of the linear chirp pump pulses enhances the modulation depth. This increases the sensitivity of the DEER experiment [56, 144, 147]. The use of 5-pulse and 7-pulse sequences also help to increase the dipolar evolution window [56, 144]."

Round 3

Reviewer 2 Report

The review has been improved and can be accepted once the following mistakes have been addressed:

l 35: „The 30% of genomes“ ?

l 50: It is not impossible for X-ray crystallography to reveal dynamic information. After all, there are many structure of transporters in various conformations. So I would say it is difficult for X-ray crystallography to reveal dynamic information

Figure 4: in C, it is still difficult to see the spin label.

l 186: Names of bacteria should be in italic.

Figure 5: The names of the mutants are missing in panel B

l388 : structural biology 

Author Response

l 35: „The 30% of genomes“ ?

That sentence has been updated as follows in the revised manuscript: (see page 1)

". In humans, 30% of the genome encodes membrane proteins [2-4]"

l 50: It is not impossible for X-ray crystallography to reveal dynamic information. After all, there are many structure of transporters in various conformations. So I would say it is difficult for X-ray crystallography to reveal dynamic information

As suggested, we have updated that sentence in the revised manuscript as follows: (see page 2)

" However, it is difficult for X-ray crystallography to reveal dynamic information of the most proteins in a membrane.

"

Figure 4: in C, it is still difficult to see the spin label.

 The Figure 4C has been updated to make the spin label more visible (see page 5).

l 186: Names of bacteria should be in italic.

As suggested, the name of the bacteria was modified to italic. (see page 6)

Figure 5: The names of the mutants are missing in panel B

The name of the mutants are included in the revised manuscript. (See Figure 5).

l388 : structural biology 

As suggested, we have updated in the revised manuscript. (see page 14)

Reviewer 3 Report

  1. The authors could also mention the disadvantages of the non-nitroxide spin labels as well as the 5-pulse DEER and the 7-pulse CP-PELDOR (for which they are not very popular). In Figure 8 and in the legend, it is written as seven-pulse DEER, which should be corrected to 7-pulse CP-PELDOR. Although the authors shows the sequence for the reverse five-pulse DEER, nothing is mentioned about in the text. 
  2. Line 346: The statement that 7-pulse CP-PELDOR allows for more substantial  increase in the length of the dipolar evolution time (as compared to 5-pulse DEER) is not fully valid as there is no systematic study on any membrane protein. 
  3. Minor spell checks are required, for e.g., italicizing E. coli in a few places.

Author Response

  1. The authors could also mention the disadvantages of the non-nitroxide spin labels as well as the 5-pulse DEER and the 7-pulse CP-PELDOR (for which they are not very popular).

As suggested, the following descriptions have been included to describe the limitations of non-nitroxide spin labels and 5-pulse and the 7-pulse CP PELDOR in the revised manuscript. Page 11,

"

There is also disadvantage using non-nitroxide spin labels. The Gd-based and trityl labels are bulkier than nitrixide spin labels that can cause perturbation in protein structure and function [56]. Hence, a care must be taken while choosing spin labeling sites to avoid these perturbations."

"

These multi pulse DEER experiments introduce echo crossing artifacts in DEER traces [144, 145, 147]. These artifacts can be minimized by using eight to thirty two-step phase cycling schemes [144].

"

  1. In Figure 8 and in the legend, it is written as seven-pulse DEER, which should be corrected to 7-pulse CP-PELDOR. Although the authors shows the sequence for the reverse five-pulse DEER, nothing is mentioned about in the text. 

As suggested, we have updated the Figure 8E and the corresponding figure legend: (see Page 13, Figure 8 and the legend). The text has been also updated to include the reverse five-pulse DEER . (see page 11).

  1. Line 346: The statement that 7-pulse CP-PELDOR allows for more substantial  increase in the length of the dipolar evolution time (as compared to 5-pulse DEER) is not fully valid as there is no systematic study on any membrane protein. 

As suggested, we have removed that statement and updated that sentence as follows in the revised manuscript (see page 11):

"

A seven-pulse Carr-Purcell PELDOR sequence with multiple pump pulses (see Figure 8E) also leads to improved sensitivity in the measurement of long range distances. [144, 145, 147].

"

  1. Minor spell checks are required, for e.g., italicizing E. coliin a few places.

We corrected as suggested by reviewers. (see page 14)